# Mechanism of Type IA Topoisomerases

**DOI:** 10.3390/molecules25204769

**Published:** 2020-10-17

**Authors:** Tumpa Dasgupta, Shomita Ferdous, Yuk-Ching Tse-Dinh

**Affiliations:** 1Department of Chemistry and Biochemistry, Florida International University, Miami, FL 33199, USA; tdasg002@fiu.edu (T.D.); sferd001@fiu.edu (S.F.); 2Biomolecular Sciences Institute, Florida International University, Miami, FL 33199, USA; 3Biochemistry PhD Program, Florida International University, Miami, FL 33199, USA

**Keywords:** topoisomerase, type IA, DNA supercoiling, RNA topology, decatenation

## Abstract

Topoisomerases in the type IA subfamily can catalyze change in topology for both DNA and RNA substrates. A type IA topoisomerase may have been present in a last universal common ancestor (LUCA) with an RNA genome. Type IA topoisomerases have since evolved to catalyze the resolution of topological barriers encountered by genomes that require the passing of nucleic acid strand(s) through a break on a single DNA or RNA strand. Here, based on available structural and biochemical data, we discuss how a type IA topoisomerase may recognize and bind single-stranded DNA or RNA to initiate its required catalytic function. Active site residues assist in the nucleophilic attack of a phosphodiester bond between two nucleotides to form a covalent intermediate with a 5′-phosphotyrosine linkage to the cleaved nucleic acid. A divalent ion interaction helps to position the 3′-hydroxyl group at the precise location required for the cleaved phosphodiester bond to be rejoined following the passage of another nucleic acid strand through the break. In addition to type IA topoisomerase structures observed by X-ray crystallography, we now have evidence from biophysical studies for the dynamic conformations that are required for type IA topoisomerases to catalyze the change in the topology of the nucleic acid substrates.

## 1. Introduction

Normal cell growth requires replication of the genome and regulated transcription. Certain enzymes play crucial roles in these vital cellular processes. For example, DNA polymerase enzymes are essential for DNA replication [1] and RNA polymerases are required for transcription [2]. Because cellular genomes exist as very long double-stranded DNA, topoisomerases are needed for their unique role as master regulators of DNA topology during DNA replication, transcription, recombination, chromosome remodeling and many other genomic processes [3,4,5,6,7]. Topoisomerases catalyze the interconversion of different topological forms of DNA by creating transient breaks on one or both strands of the DNA duplex [8,9]. The twin supercoiling domain model proposed by Liu and Wang described the topological problems encountered during transcription that would require topoisomerase action ahead and behind the elongating RNA polymerase complex [10,11]. In bacteria, topoisomerase I belonging to the type IA subfamily relaxes the negative supercoiling generated by transcription behind the RNA polymerase complex to prevent the formation of DNA–RNA hybrids/R-loops that in turn can inhibit the transcription process [6,10,12,13,14]. In addition to the relaxation of negatively supercoiled DNA and preventing hypernegative supercoiling, type IA topoisomerases can also catalyze the decatenation of replication intermediates [15,16,17,18,19] and the knotting or unknotting of single-stranded DNA circles or nicked duplex DNA [20,21]. Every living organism has at least one type IA topoisomerase that can resolve topological barriers, including replication and recombination intermediates or other entangled DNA structures, by passing DNA through a transient break in a single strand of DNA [7]. The ability to unknot single-stranded RNA circles was first observed for *Escherichia coli* topoisomerase III [22]. This RNA topoisomerase activity might have been related to the similarity between *E. coli* topoisomerase III (TOP3) and an ancestral type IA topoisomerase present in the RNA world. The interest in the physiological significance of RNA topoisomerase activity was greatly heightened by the findings that human topoisomerase IIIβ (TOP3B) has both DNA and RNA topoisomerase activities [23,24]. Furthermore, mutations in human TOP3B are linked to disorders in neurodevelopment and mental health [23,24,25,26,27]. Many other type IA topoisomerases in all three domains of life have been found to possess RNA topoisomerase activity [28,29,30]. While TOP3B is likely to participate in the regulation of transcription-associated supercoiling and R-loops within the nucleus [31,32,33,34], TOP3B has been shown to bind to mRNA within the cytoplasm [23,24,25] and may play a role in translation. Potential roles of TOP3B in unlinking molecules of mRNA or overcoming torsional stress of mRNA have been proposed [29], but the exact cellular functions of the RNA topoisomerase activity remain to be fully elucidated. Remarkably, TOP3B has recently been demonstrated to be a host factor hijacked for the efficient viral replication of positive-sense single-stranded RNA viruses that include flaviviruses and coronaviruses [35]. Figure 1 shows some of the potential topological barriers occurring during the life cycle of a positive-sense single-stranded RNA virus that may require the RNA topoisomerase activity of human TOP3B. TDRD3 plays an important role in the stabilization and activation of the DNA and RNA topoisomerase activities of human and Drosophila TOP3B [28,31,36]. CRISPR/Cas9-based deletion that targeted the TOP3B–TDRD3 complex did not affect *Flavivirus* translation and replication, but was found to diminish the late-stage production of infectious virus particles [37]. It is not known yet if the TOP3B–TDRD3 complex is involved similarly in the different stages of the coronavirus life cycle.

Two recent reviews on type IA topoisomerases have discussed the essential functions of bacterial type IA topoisomerase [14] and the many versatile collaborations engaged by eukaryotic TOP3 to carry out different topological transactions [38]. This review on the mechanism of type IA topoisomerases will focus more on the results from recent structural, biophysical, biochemical and genetic studies that help us gain a better understanding of how type IA topoisomerases can act as magicians to manipulate the topology of both DNA and RNA, in addition to key remaining questions on the catalytic mechanism.

## 2. General Classification

Topoisomerases are classified into two major types, type I and type II topoisomerases, based on their ability to cleave one or both DNA strands, respectively, during the catalytic process. According to structural homologies and reaction mechanisms, type II topoisomerases can be subdivided into two subgroups: type IIA (including gyrase and topoisomerase IV found in prokaryotes; topoisomerase IIα and topoisomerase IIβ in humans) and type IIB (topoisomerase VI and VIII found in archaea and bacteria) [39,40,41]. Type I topoisomerases can be divided into type IA and type IB, IC subgroups [41]. Type IA topoisomerases, ubiquitous in bacteria, archaea and eukarya [41], can relax negatively supercoiled DNA but not positive supercoils because of the requirement of single-stranded DNA for binding [42]. Type IA topoisomerase introduces a transient cleavage of the single DNA strand and forms a covalent linkage to the 5′-terminal phosphate of the cleaved DNA, followed by the passing of an intact strand (T-strand) of DNA through the DNA break in the cleaved G-strand and the subsequent religation of the nicked G-strand of DNA. This process is referred to as the enzyme-bridged or enzyme-gated mechanism [21,43,44]. Type IB and type IC topoisomerases form a covalent linkage to the 3′-terminal phosphate during catalysis that involves a 360° rotation of the cleaved free end of DNA around the intact strand to relax both positive and negative supercoils with the controlled swivel or controlled rotation mechanism [4,45,46]. Type IC differs from type IB or any other known topoisomerase in sequence or structure, with a unique fold in its N-terminal domain [47].

## 3. N-Terminal Type IA Core Domains—Binding and Positioning of the G-strand for Cleavage at the Active Site

The first information of type IA topoisomerase structure came from the crystal structure of the 67 KDa N-terminal fragment of *E. coli* topoisomerase I (PDB 1ECL) [44]. We can see in this structure a type IA core domain structure formed by domains D1–D4 that is present in the N-terminal region of all type IA topoisomerase I and topoisomerase III (Figure 2) determined subsequently for *E. coli* topoisomerase III (PDB 1D6M) [48], *Thermotoga maritima* topoisomerase I (PDB 2GAI) [49], *Mycobacterium tuberculosis* topoisomerase I (PDB 5D5H) [50], *Streptococcus mutans* topoisomerase I (PDB 6OZW) [51], *Mycobacterium smegmatis* topoisomerase I (PDB 6PCM) [52], human topoisomerase III alpha (PDB 4CGY) [53] and III beta (PDB 5GVC) [54]. In the structure of reverse gyrases from *Archaeoglobus fulgidus* (PDB 1GKU) [55] and *T. maritima* (PDB 4DDU) [56], type IA core domains are found in the C-terminal region preceded by the helicase-like domain [55]. The elucidation of the crystal structure of *E. coli* topoisomerase I core domains [44] provided support and further detail for the previously proposed enzyme-gated mechanism of catalysis that should be applicable for the other members of the type IA subfamily of topoisomerases. The interior cavity of the torus structure observed in this study was noted to be big enough to hold single- and double-stranded DNA substrates to accomplish the relaxation of supercoiled DNA or catenation/decatenation of nicked double-stranded DNA. The G-strand of DNA fits into a binding groove in domain D4, and follows the path of one strand of a B form DNA, as observed in the subsequently obtained cocrystals of *E. coli* topoisomerase III (PDB 1I7D) [57] and topoisomerase I (PDB 1MW8) [58]. The active site tyrosine is part of domain D3 and situated at the interface of domains D1 and D3 and responsible for the transient breakage of the G-strand of DNA along with the subsequent formation of the phosphoryl-tyrosyl covalent complex (Figure 3). The D111N mutation of *E. coli* topoisomerase I was found to be extremely lethal because the resulting deficiency in DNA religation leads to the accumulation of the covalent intermediate [59]. This mutation was utilized to obtain the crystal structure of the covalent complex formed between *E. coli* topoisomerase I core domains and cleaved DNA (PDB 3PX7) [60]. While the conformation of the individual N-terminal domains remained largely the same in this covalent complex and in the full-length *E. coli* topoisomerase I structure with the C-terminal domains present (PDB 4RUL) [61], there are conformational changes in the N-terminal core domains including the relative orientations of the individual domains that allow type IA topoisomerases to bind the G-strand and form the active site (Figure 3A) as observed for *E. coli* topoisomerase I, *E. coli* topoisomerase III and *M. tuberculosis* topoisomerase I [57,58,60,62].

A comparison between the structures of the topoisomerase–DNA complex and apoenzyme showed that conformational change in domain D4 created the binding groove for the G-strand of DNA (Figure 3B). This is brought about by the movement of an alpha helix that follows a strictly conserved glycine residue (Gly194 in *E. coli* topoisomerase I, Figure 3D). The flexibility of this glycine may facilitate this conformational change required for G-strand binding [63]. Polar and positively charged residues in this alpha helix, including strictly conserved Arg195 and Gln197 in *E. coli* topoisomerase I and Arg194 and Gln196 of *M. tuberculosis* topoisomerase I, have been noted to interact with the G-strand phosphodiester backbone [57,60,62]. Site-directed mutagenesis of *E. coli* topoisomerase at Gly194, Arg195 and Gln197 confirmed that mutations at these residues reduced the relaxation activity significantly [63,64]. An additional strictly conserved arginine residue in domain D4 (corresponding to Arg507 in *E. coli* topoisomerase I, Figure 3D) also interacts with a phosphate of the G-strand [60,62].

The deoxyribose and bases of the G-strand interact with other polar, positively charged and aromatic residues that extend from the interface of core domains D1 and D3 into D4 (Figure 3B). Arg168 and Asp172 in *E. coli* topoisomerase I are strictly conserved in type IA topoisomerases and have been shown with site-directed mutagenesis to be required for the relaxation and G-strand DNA cleavage [60,64]. The R168C substitution in *E. coli* topoisomerase I was found recently in genetic studies to increase the rate of sequence deletion and duplication events, resulting in a mutator phenotype [65]. Another *E. coli* topoisomerase I mutation, R35P, was associated with an increase in short-sequence deletions in the same study. It is not known if the genetic instability is due to the effect of the reduction in topoisomerase I relaxation activity on DNA supercoiling, or the disruption of the enzyme catalytic cycle by the mutation [65].

Topoisomerase I and reverse gyrase enzymes in the type IA topoisomerase family exhibit a preference for a cytosine base at four nucleotides upstream (−4) of the DNA cleavage sites [66,67,68]. In the structure of the *E. coli* topoisomerase I covalent complex and *M. tuberculosis* topoisomerase I noncovalent complex representing the pretransition state (PDB 6CQ1), this cytosine base fits into a sterically restrictive cavity formed by residues that are conserved in the topoisomerase I and reverse gyrase sequences (highlighted in blue in Figure 3D). In these crystal structures, a tyrosine side chain (Tyr177 in *E. coli* topoisomerase I) wedges between the −4 and −5 DNA bases and creates a kink in the G-strand. The substitution of Tyr177 with serine to abolish the base-stacking interaction resulted in the complete loss of DNA cleavage and relaxation activity [69]. Phenylalanine at this position (Figure 3D) likely plays the same role in reverse gyrases. Alanine substitution at Arg169 switched the DNA cleavage sequence site preference to having an adenine at the −4 position instead of cytosine and reduced the relaxation activity by 150-fold, while substitution at Arg173 reduced DNA cleavage and relaxation activity less severely, and did not change the cytosine preference [69]. The specific binding of the cytosine positions the phosphate four nucleotides downstream at the position for nucleophilic attack by the active site tyrosine. Topoisomerase III sequences have different amino acid residues present at positions corresponding to Arg169, Arg173 and Tyr177 in the alignment (Figure 3D), and may have a different mechanistic basis for selectivity in cleavage sites.

For RNA topoisomerase activity, the G-strand is expected to be accommodated in the same binding groove in type IA topoisomerase core domains. The helix in D4 that follows the flexible glycine may play a similar role in movement to facilitate G-strand binding and interact with the phosphodiester backbone of the G-strand RNA. Structural and sequence variation in the binding groove may influence the relative efficiency of DNA versus RNA topoisomerase activity. Sequence selectivity of RNA cleavage by type IA topoisomerases have not been analyzed. Alanine substitution at Arg173 of *E. coli* topoisomerase I was found to have a more severe effect on the enzyme’s RNA unknotting activity than its DNA unknotting activity, suggesting that the cytosine may need to be bound selectively at the −4 position for the RNA G-strand cleavage by *E. coli* topoisomerase I [70]. In addition to the preference of a cytosine at the −4 position, there may also be structural features that are relevant for the binding and cleavage of RNA as the G-strand due to the presence of the 2′-hydroxyl groups on the RNA ribose rings.

## 4. Mechanism of the G-strand Cleavage and Religation

Type IA topoisomerases achieve the relaxation of supercoiled DNA through a transesterification reaction that includes two sequential nucleophilic attacks involving the active site tyrosine residue situated in D3 and positioned at the junction of domains D1 and D3 [44]. In type IA topoisomerases, the hydroxyl group in the side chain of the active site tyrosine residue is responsible for the first nucleophilic attack on the scissile phosphate of a single-stranded DNA resulting in the cleavage of the G-strand and the formation of the transient 5′-phospho-tyrosyl covalent linkage [66]. While the cleaved G-strand segment with the 5′ phosphate is covalently linked to the enzyme, the cleaved G-strand segment with the leaving 3′ hydroxyl is bound by multiple noncovalent interactions [60]. The covalent and noncovalent interactions on both sides of the G-strand cleavage site not only facilitate the enzyme-bridged T-strand passage through the break, but also prevent any inadvertent release of the cleaved DNA that can harm the chromosome integrity. After the intact T-strand passes through the nick, rejoining of the G-DNA backbone occurs through the second nucleophilic attack by the 3′-OH group of the cleaved G-strand on the phosphotyrosine linkage to release the enzyme from the covalent complex and allow the enzyme to release the substrate and be ready for the next catalytic cycle [71,72].

A number of conserved amino acids with acidic and basic side chains are present in close proximity of active site tyrosine in type IA topoisomerases to play a role in G-strand cleavage and religation [73,74]. A water molecule has been postulated to be acting as the general base for deprotonation of the tyrosine hydroxyl nucleophile for DNA cleavage [57,62]. The deprotonation of the nucleophile by water is likely to be in concert with the formation of the transition state at the scissile phosphate, similar to the nucleotide transfer mechanism proposed for DNA polymerases [75]. The positively charged side chain of the strictly conserved arginine residue separated by one residue from the active site tyrosine (Arg321 in *E. coli* topoisomerase I, Figure 3C) plays a critical role in stabilizing the negative charge on the hydroxyl nucleophile and the transition state. The positive charge of divalent ions can also activate the nucleophile and stabilize the transition state for both G-strand cleavage and religation even though divalent ions are absolutely required only for G-strand religation, and can be absent for cleavage of the G-strand by type IA topoisomerases. However, when Arg321 in *E. coli* topoisomerase I or the corresponding Arg327 in *Yersinia pestis* topoisomerase I was substituted with an aromatic residue, DNA cleavage could still take place but became divalent ion dependent [76]. Moreover, the presence of divalent ions cannot compensate for the loss of this arginine residue in G-strand religation, resulting in a dominant lethal cell killing because of the accumulation of topoisomerase I-mediated DNA breaks [76]. A R338W mutation introduced at the corresponding arginine residue in human topoisomerase IIIβ has been shown to facilitate trapping of the intracellular covalent complex with both DNA and RNA [77], confirming a similar role for this arginine residue in the cleavage and rejoining of the RNA G-strand.

Type IA and type IIA topoisomerases require divalent ions for catalytic activity. At the active sites of these topoisomerases, divalent ions are coordinated by the TOPRIM motifs (with a conserved glutamate and two conserved aspartates DxD) also seen in many nucleotidyl transferases/hydrolases [78]. In the crystal structure of *M. tuberculosis* topoisomerase I [62], one Mg^2+^ ion is coordinated with the negatively charged carboxylate side chain of Glu24 (corresponding to Glu9 in *E. coli* topoisomerase I, Figure 3A) and Glu111 directly, and to Glu113 indirectly through a water molecule, in addition to the scissile phosphate. The positive charge of Mg^2+^ can help to position the 3′-OH of the cleaved G-strand for a nucleophilic attack on the phosphotyrosine linkage and stabilize the transition state for DNA rejoining. The DNA cleavage also became Mg^2+^ dependent when the first aspartate of the DxD TOPRIM motif was mutated to asparagine without the negatively charged side chain [59,79]. Overexpression of the D111N mutant topoisomerase I is toxic to the cells because of deficiency in DNA religation resulting in the accumulation of the cleavage complex on chromosomal DNA. The D111N mutation in the *E. coli* topoisomerase I N-terminal core domain fragment enabled the isolation of the covalent complex formed with cleaved oligonucleotide for structural determination [60]. Other mutations that affect Mg^2+^ binding at the active site of bacterial topoisomerase I also resulted in bacterial cell death when the mutant topoisomerase I was overexpressed because of the inhibition of DNA religation and accumulation of DNA breaks [79,80,81,82]. The requirement for DNA religation is more stringent than for DNA cleavage because both the phosphotyrosine linkage and 3′-hydroxyl nucleophile have to be placed exactly at positions required for DNA rejoining to take place.

In the crystal structures of human topoisomerase IIIα [53] and human topoisomerase IIIβ [54], DNA substrate is not present. A single Mg^2+^ is bound directly to the Glu side chain of the TOPRIM motif and through a water molecule to the first Asp of the TOPRIM DxD. It is expected that upon binding of the G-strand at the active site, the scissile phosphate will displace the water molecules seen in the crystal structure as ligands for the Mg^2+^. It cannot be ruled out that a transient interaction with additional Mg^2+^ not currently observed in the crystal structures of type IA topoisomerases can take place during the course of DNA cleavage and rejoining. 

In addition to acting as a ligand for Mg^2+^, the TOPRIM glutamate side chain has been proposed to interact directly during DNA cleavage as a general acid [57] with the G-strand 3′-hydroxyl leaving group to provide a proton from a nearby positively charged histidine (His365 in *E. coli* topoisomerase I) side chain via proton relay through the D111 side chain [62,83]. Reversal of the proton relay may take place during religation for the glutamate to act as a general base in the activation of the 3′-hydroxyl nucleophile [60]. Mutation of the TOPRIM glutamate to alanine or glutamine abolished the DNA cleavage and relaxation activity [62] but did not affect the RNA cleavage activity observed for *M. smegmatis* topoisomerase I, suggesting that the 2′-OH of RNA could potentially participate in the proton relay for the RNA cleavage [30].

## 5. C-Terminal Domains—Binding of T-strand

Though the type IA topoisomerase core domain that forms the characteristic torus structure contains all the highly conserved motifs responsible for G-strand binding and cleavage religation, the C-terminal domains of bacterial topoisomerase I have been shown to be required for removing negative supercoils from DNA rapidly in a processive mechanism [61,72,84,85,86,87]. Results from these studies indicated that the C-terminal domain of type IA topoisomerases possesses a significant affinity to substrate DNA. Unlike the N-terminal core domains, the C-terminal domains among type IA topoisomerases greatly varied in size and sequence. Two distinct types of structural motifs (Figure 4A) have been observed in the crystal structures of bacterial topoisomerase I [28,49,50,61].

Tetracysteine motifs that form the zinc ribbon fold [88] can be found in the topoisomerase I gene of a majority of bacterial species. The zinc ribbon fold comprises a four-stranded antiparallel β-sheet with a Zn(II)-binding site on the top of the motif (Figure 4A). The crystal structure of full-length *E. coli* topoisomerase I [61] showed three such Topo_C_ZnRpt zinc ribbon domains (D5-D7) followed by two zinc ribbon-like domains (D8, D9) that did not bind to Zn(II) due to the absence of cysteines [89]. The four cysteines in the single zinc ribbon domain (D5) in *T. maritima* topoisomerase I did not bind to the Zn(II) ion in the crystal structure but formed two disulfide bonds to stabilize the zinc ribbon fold [49]. While the deletion of the C-terminal domains in *E coli* topoisomerase I resulted in the complete loss of relaxation activity [90], *T. maritima* topoisomerase I retained some activity upon deletion of the C-terminal domain [91].

In certain species of bacteria in the Actinobacteria phylum including members of the Mycobacterium and Streptomyces genera, the C-terminal domains are organized with repeated Topo_C_Rpt domains first predicted based on the *M. tuberculosis* topoisomerase I crystal structure [50] that do not have Zn(II)-binding cysteines, ending with a tail rich in positively charged lysines and arginines. The Topo_C_Rpt fold has an antiparallel four-stranded β-sheet flanked by a C-terminal helix on one side (Figure 4A).

In the crystal structures of the topoisomerase–DNA complex, π–π stacking interactions between tyrosine and phenylalanine amino acids at specific positions of the Topo_C_ZnRpt or Topo_C_Rpt motifs and nucleotide bases are utilized for DNA binding by the C-terminal domains of *E. coli* and *M. smegmatis* topoisomerase I (Figure 4B). Additional interactions with the ssDNA include hydrogen bonds and cation–π interactions. Flexible loops connect between these C-terminal domains and to the N-terminal core domains. Topoisomerase IIIα and IIIβ in higher eukaryotes have large numbers of cysteines in their C-terminal domains that could potentially form multiple Zn(II)-binding motifs that should be similar to the Topo_C_ZnRpt motifs in *E. coli* topoisomerase I. Structural studies are needed to determine the exact folding of these C-terminal motifs in eukaryotic topoisomerase III enzymes. In Actinobacteria such as Mycobacteria and Streptomyces, a lysine-rich C-terminal tail follows the Topo_C_Rpt motifs to also participate in DNA binding [84,86]. Biochemical analysis demonstrated that these elements in type IA topoisomerases contribute to the interaction with the single-stranded DNA region in the substrate and are important for the processivity of enzyme activity [86,87,92,93,94]. There is further evidence that the bacterial topoisomerase I C-terminal domains have a specific role for interacting with the T-strand in strand passage for the efficient recognition and relaxation of negatively supercoiled DNA [52,61,86,95]. Figure 5 illustrates a model of relaxation of supercoiled DNA by bacterial topoisomerase I that adds the binding of T-strand by C-terminal domains to the first step of previously proposed topoisomerase I catalytic mechanism [72].

An RGG-box rich in arginines and glycines follows the Zn(II)-binding motifs in the C-terminal domains of topoisomerase IIIβ found in higher eukaryotes [24,96]. A similar RGG-box is found in many proteins involved in mRNA processing [97]. The deletion of the RGG-box from topoisomerase IIIβ reduced both the DNA and RNA topoisomerase activities [24]. The RGG-box in topoisomerase IIIβ may contribute to both DNA and RNA binding. Post-translational methylation of specific arginine residues in the topoisomerase IIIβ RGG-box enhances the DNA and RNA topoisomerase activities, and also the interaction between topoisomerase IIIβ and its auxiliary factor TDRD3 [32]. The acetylation of lysine residues can also influence the topoisomerase activity of *E. coli* topoisomerase I [98,99].

It can be postulated that the last universal common ancestor (LUCA) of type IA topoisomerase may resemble the bacterial and archaeal topoisomerase III enzymes consisting of the N-terminal core domains and relatively short C-terminal domain. The C-terminal domains observed in the topoisomerase I of bacteria and the topoisomerase III of higher eukaryotes have evolved to enhance the interactions with nucleic acid substrates and protein–protein interactions using the repeated C-terminal domains and basic amino acid residues. The protein–protein interactions with other cellular proteins are relevant for the physiological functions and regulation of type IA topoisomerase activities [38]. Basic residues in the C-terminal domains of *E. coli* topoisomerase I [100] and the C-terminal tail of *M. smegmatis* topoisomerase I [101] have been proposed to interact directly with the β′ subunit of RNA polymerase for the function of topoisomerase I in transcription elongation [14,102,103] to suppress hypernegative supercoiling and R-loop accumulation [14,104,105].

A recent report of an additional active site in the C-terminal domains of *Helicobacter pylori* topoisomerase I [106] further illustrates the diverse properties of type IA topoisomerase C-terminal domains. There are eight tyrosine residues present in the C-terminal domains of *H. pylori* topoisomerase. It is not known which of these tyrosine residues may act as an alternative active site nucleophile for relaxation of negatively supercoiled DNA.

## 6. Opening and Closing of the G-strand Gate for Strand Passage

In the enzyme-bridged mechanism [21], the two ends of the cleaved G-strand in the covalent intermediate formed by type IA topoisomerases need to be separated by a significant distance through an extensive conformational change of the covalent complex to create the opening at the gate for strand passage. Significant conformational changes of bacterial topoisomerase I have been detected by bulk fluorescence measurements [107,108]. Trapped ion mobility spectrometry–mass spectrometry (TIMS–MS) also demonstrated microheterogeneity of *E. coli* topoisomerase I conformational states [109]. The extent of gate opening was measured experimentally for *E. coli* topoisomerase I and topoisomerase III in single-molecule assays [110]. The distance between the cleaved ends of single-stranded DNA bound covalently to the topoisomerases was found to increase by as much as 6 nm. Such a distance was estimated by molecular dynamics simulation to be required for the passage of a double-stranded DNA through the break for a catenation/decatenation reaction involving double-stranded circular DNA with a nick or single-stranded gap [110]. A significant opening of the G-strand gate would also be needed for the passage of a single strand of DNA through the break to catalyze the relaxation of negatively supercoiled DNA.

According to the model for type IA topoisomerase catalysis proposed when the crystal structure of the core domains first showed [44] a toroid enclosed by interactions of D3 with D1 and D4, D3 must move away from D1 and D4 for gate opening. Following the entry of DNA into the interior of the toroid hole, D3 needs to be brought back by a conformational change to close the G-strand gate and religate the break in the G-strand (Figure 5). Opening of the toroid hole also needs to occur at the end of the catalytic cycle for the release of DNA from the interior.

A decatenation loop present in *E. coli* topoisomerase III but not topoisomerase I has been proposed to keep the gate open to assist the catalysis of decatenation [111]. In contrast, the rapid closing of the DNA gate formed by *E. coli* topoisomerase I observed in the single-molecule assays would allow a fast rate of relaxation of negatively supercoiled DNA [110]. The hinge region between domains D2 and D4 could play an important role in the gate opening and closing. Combined techniques of magnetic tweezers and total internal reflection fluorescence microscopy also detected *E. coli* topoisomerase I conformational change that is necessary for strand passage, but an alternative model of sliding the domains past each other to create a gate for capturing the T-strand of DNA into the interior of the toroid was proposed [112].

## 7. Key Remaining Questions

The mechanism of the G-strand gate opening and T-strand transport by type IA topoisomerases remains to be fully elucidated. Communication between the N-terminal core domains and the C-terminal domains bound to the T-strand may be required for coordinating the opening and closing of the G-gate with strand passage. The guiding of the T-strand in and out of the toroid hole is also not well understood. Direct evidence and characterization of additional transient intermediates that are part of the proposed catalytic cycle would further our understanding of the mechanism at the molecular level.

The recent discovery of the RNA topoisomerase activity for type IA topoisomerases expanded the scope of potential cellular functions for this ubiquitous class of topoisomerases. Even though the interaction between topoisomerase IIIβ and mRNA in vivo has been linked to synaptic functions [23,24,25], there is no direct experimental evidence so far for a change in cellular RNA topology catalyzed by RNA topoisomerase activity. Separation of function mutations or inhibitors that can be identified for topoisomerase IIIβ would be valuable research tools for investigating the cellular activity and function of topoisomerase IIIβ.

## Figures and Tables

**Figure 1 molecules-25-04769-f001:**
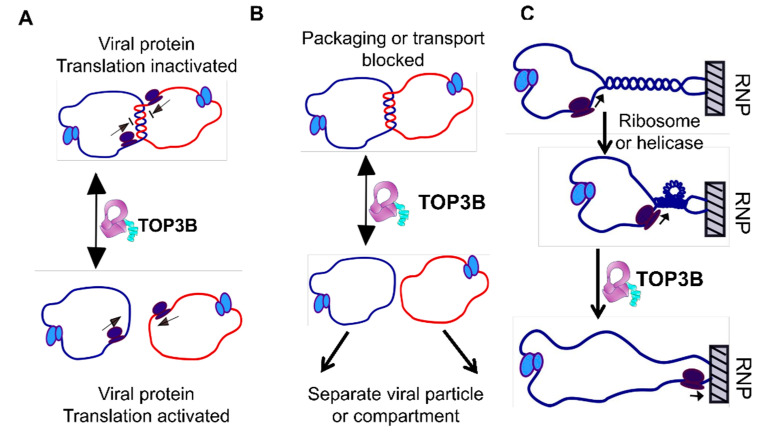
Some of the potential topological problems during the life cycle of a positive-sense single-stranded RNA virus that may require the RNA topoisomerase activity of human TOP3B. Genome circularization can be mediated by RNA–RNA interactions and proteins binding to the 5′ and 3′ ends. Decatenation of the catenated circular viral genome by TOP3B is required to remove blocks of (**A**) viral protein translation, (**B**) viral RNA transport and packaging. (**C**) When a translating ribosome or a helicase unwinds a duplex region in a viral RNA hairpin, and if the hairpin is bound to an immobile RNP or cellular matrix, the helical torsion will need to be relaxed by TOP3B. The figure is modified from a published version for potential TOP3B action on mRNA in [29].

**Figure 2 molecules-25-04769-f002:**
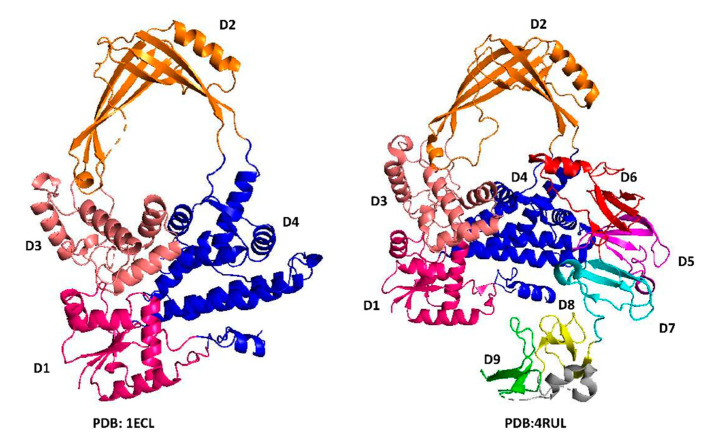
Domain arrangement in *E. coli* topoisomerase I as seen in the crystal structure of the N-terminal core domains (PDB 1ECL) or full-length enzyme (PDB 4RUL). D1: amino acids 4–36, 81–157 colored in pink; D2: amino acids 216–278,405–472 colored in orange; D3: amino acids 279–404 colored in salmon; D4: amino acids 61–80,158–215,473–590 colored in blue; D5: amino acids 591–635 colored in magenta; D6: amino acids 636–706 colored in red; D7: amino acids 707–754 colored in cyan; D8: amino acids 755–795 colored in yellow; helix hairpin linker: amino acids 796–824 colored in gray; D9: amino acids 825–865 colored in green.

**Figure 3 molecules-25-04769-f003:**
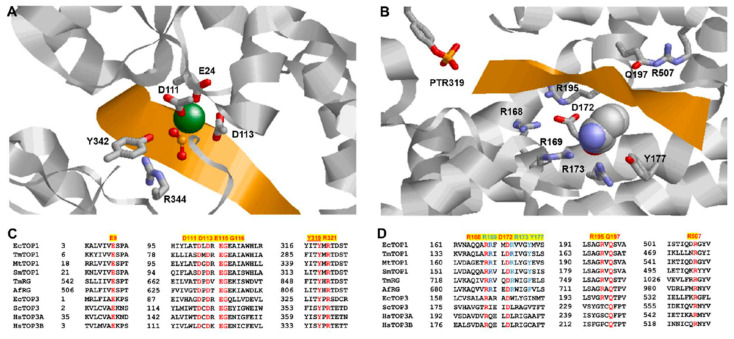
Type IA topoisomerase core domains interactions with the G-strand. (**A**) Active site of the *M. tuberculosis* topoisomerase I noncovalent complex (PDB 6CQ2) showing an interaction between the catalytic tyrosine and adjacent arginine with the scissile phosphate of the G-strand (in gold). A Mg^2+^ ion (in green) interacts with the scissile phosphate and TOPRIM acidic residues. (**B**) Structure of the *E. coli* topoisomerase I covalent complex (PDB 3PX7) showing the positioning of the G-strand by residues interacting with the G-strand backbone and a cytosine base (shown in the space-filling display) at a distance of four nucleotides from the phosphoryl tyrosine (PTR) formed from the cleavage of the G-strand. (**C**) Alignment of residues strictly conserved at the active site and type IA topoisomerases including bacterial topoisomerase I, reverse gyrase (RG) and topoisomerase III from prokaryotes and eukaryotes. Species represented include Ec: *E. coli*, Tm: *Thermotoga maritima*. Mt: *Mycobacterium tuberculosis*, Sm: *Streptococcus mutans*, Af: *Archaeoglobus fulgidus*, Sc: *Saccharomyces cerevisiae*, Hs: *Homo sapiens.* (**D**) Alignment of residues (shown in red) that are strictly conserved in type IA topoisomerases for interacting with the G-strand backbone. Residues conserved for the specific binding of a cytosine base 4 nucleotides upstream of the scissile phosphate are shown in blue.

**Figure 4 molecules-25-04769-f004:**
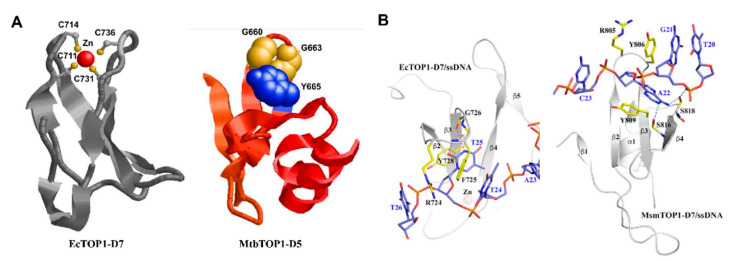
Structural motifs found in the C-terminal domains of bacterial topoisomerase I. (**A**) The Topo_C_ZnRpt motif with the Zn(II) ion coordinated by four cysteines in *E. coli* topoisomerase I (from PDB 4RUL) and Topo_C_Rpt motif with conserved GxxGPY residues (in the space-filling display) in *M. tuberculosis* topoisomerase I (from PDB 5UJ1). (**B**) Comparison of ssDNA binding by the C-terminal domain of *E. coli* topoisomerase I and *M. smegmatis* topoisomerase I (from the Supplementary Information of [52]). Conserved aromatic residues from each C-terminal domain form π–π stackings with the nucleotide bases.

**Figure 5 molecules-25-04769-f005:**
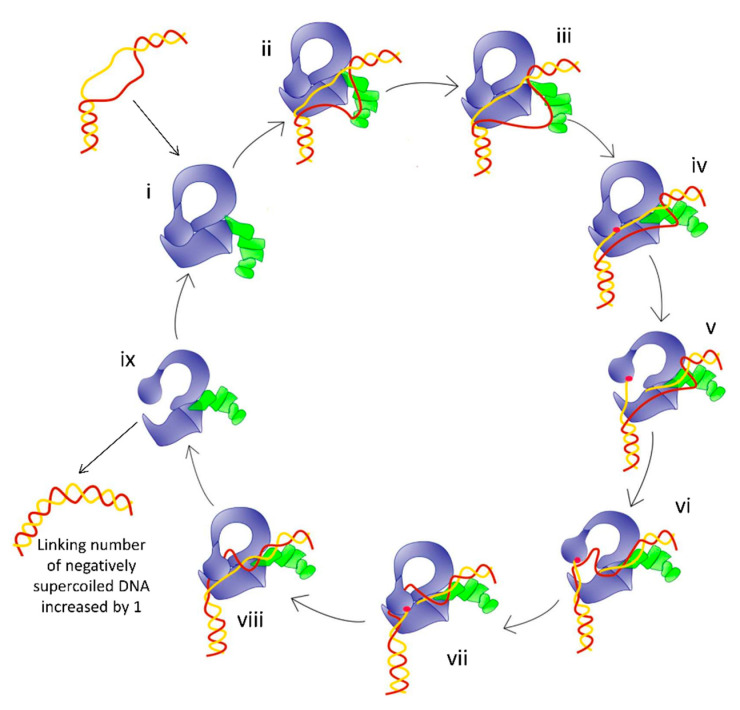
Model for the relaxation of supercoiled DNA by bacterial topoisomerase I based on the crystal structures of *E. coli* and Mycobacteria topoisomerase I. (i) Apo enzyme; (ii) C-terminal domains (green) bind ssDNA as T-strand (red); (iii) ssDNA or G-strand (yellow) binds the N-terminal domains (blue); (iv) Active site tyrosine (red circle) becomes accessible; (v) Cleavage of the G-strand and gate opening; (vi) Passage of T-strand inside the toroid; (vii) Gate closing and trapping of T-strand; (viii) Religation of the G-strand; (ix) Gate opening and release of dsDNA.

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
