# Peer review of "Mechanism of Type IA Topoisomerases"

_molecules, 2020, doi:10.3390/molecules25204769_

Round 1

Reviewer 1 Report

This review paper ‘Mechanism of Type IA Topoisomerases’ by Tse-Dinh discusses how a type IA topoisomerase recognize and bind single-stranded DNA or RNA to initiate its required catalytic function.
The manuscript is clearly written and well summarized. I believe that the manuscript can be accepted after minor comments below are addressed.

Minor comments:
(1)
Some typo in the manuscript should be corrected. Please check all typos again in your manuscript.
For example,
In lines 208-209,
In type IA topoisomerases, The hydroxyl group…
should be corrected to
In type IA topoisomerases, the hydroxyl group…

line 228,
the active site tyrosine( (Arg321 in E. coli topoisomerase I, Figure 3C) plays…
should be corrected to
the active site tyrosine (Arg321 in E. coli topoisomerase I, Figure 3C) plays…

(2)
The resolution of figures (fig. 2, 3, 4) should be improved.

Author Response

We thank the reviewer for the comments and suggestions.  The manuscript has been carefully revised to check and correct typos and other errors.  High resolution images of the figures are provided in the revision.

Reviewer 2 Report

The manuscript from Dasgupta et al. is a thorough and well written review of the literature describing the mechanism of Type IA Topoisomerases. After clearly outlining the importance of Type IA topoisomerases in living cells, it describes on a structural level how they are able to resolve DNA topological problems. The manuscript systematically explains each step of DNA strand capture, cleavage, passage and religation, and distinguishes how these processes differ between organisms. It is an excellent review which I can recommend for publication without any major changes. It have listed some minor suggestions and spelling errors that can be considered and corrected:

Line 30: “master regulators”

Line 41: “free living organisms” is an odd term, wouldn’t “living organisms” be sufficient?

Line 89: “subdivided into type IA, IB and IC subgroups”

Line 96: Perhaps a sentence could be included to explain what distinguishes types IB and IC from each other

Lines 208-209: “topoisomerases, the hydroxyl”

Line 233: “topoisomerase I was”

Line 277: “core domains that” OR “that forms” & “structure contains”

Author Response

We thank the reviewer for the comments.  The manuscript has been revised to incorporate the suggestions and correct the errors.